# Investigating and Predicting Intentions to Continue Using Mobile Payment Platforms after the COVID-19 Pandemic: An Empirical Study among Retailers in India

Rabindra Kumar Jena





Institute of Management Technology, Nagpur 440013, India; rkjena@imtnag.ac.in

**Abstract:** The Indian retail industry has registered tremendous growth recently. The sudden emergence of Coronavirus Disease 2019 (COVID-19) and the related measures that were taken by the authorities to curb the pandemic have compelled retailers and their consumers to transact using digital platforms. This study investigates the critical precursors to retailers' behavioral intention to use mobile platforms for their business transactions in the post-pandemic era. This study adopted a framework that combined the theory of planned behavior (TPB) and self-determination theory (SDT) to predict behavioral intentions. A hybrid approach combining partial least squares structural equation modeling (PLS-SEM) and artificial neural network (ANN) techniques was used to test and validate the proposed framework. Four hundred and ninety-six participants from different central Indian cities participated in the study. PLS-SEM results confirmed that the motivational factors (need satisfaction [NS] and need frustration [NF]) significantly influence the attitude (AT), subjective norms (SN), perceived behavioral control (PBC), and behavioral intention (BI). Furthermore, the findings also established the partial mediating effect of AT, SN, and PBC on the relationship between motivational construct (NS and NF) and BI. Finally, the relationship established by SEM was successfully validated by ANN in the existence of a nonlinear relationship in the data. The findings may help retail stakeholders to support retail owners in their pursuit to continue using mobile payment systems in the post-COVID-19 world.

**Keywords:** retailer; usage intention; PLS-SEM; ANN; TPB; SDT; COVID-19; India

## 1. Introduction

India is predominantly a cash-based economy that is heavily dependent on paper-based monetary transactions (Singh et al. 2020). This status quo changed when the sudden onset of the COVID-19 pandemic enforced social distancing, lockdowns, and other measures prioritizing the adoption of a cashless economy (Singh et al. 2020). The demand for cashless payments over cash was driven by greater convenience, favorable government policies, and evolving consumer behavior. Therefore, digital payment platforms witnessed an exponential spike in the number of users during the lockdown period in India. To take advantage of the situation, retail business owners started adopting the online-merge-offline (OMO) model by integrating online and offline payments, orders, deliveries, etc. The OMO model emphasizes establishing a new business model that supports new-normal consumers. At the same time, COVID-19 has also changed banking. Consumers have formed new habits and mobile banking has become the norm. Rogers (2016) highlighted that mobile payments could positively impact business strategy. The use of mobile payments allows customers to increase the speed and convenience of payment. Mobile payments also make it possible to customize promotions and facilitate financial inclusion (Raman and Aashish 2021; Tam and Oliveira 2017), offering consumers a more satisfying and dynamic shopping experience (Chakraborty et al. 2022; Wang 2022). Mobile payment systems also help to improve the consumer experience by facilitating innovative and user-friendly purchase

processes (Leong et al. 2022). The increase in the use of digital platforms like mobile payment systems has also been driven by consumer behavior change due to the restriction of physical contact (Hoe 2020; Mora 2020).

Due to the advantages and opportunities offered by digital payment systems described above, retailers are now interested in using mobile payment applications that help them to track and improve their financial health (Chopdar et al. 2022; Sinha and Singh 2022). However, the digital payment market is dominated by a few e-commerce giants (Amazon, Alibaba, Snapdeal, etc.). The involvement of small retailers is minimal, despite the favorable environment for mobile payment in India. This is mainly due to such retailers' educational backgrounds, awareness, lack of cognitive intentions, psychological needs, fears of using mobile payment applications, etc. Meanwhile, research on mobile payment adoption is still in its infancy. The majority of the existing studies have focused on countries or multinational economic alliances like China, the USA, and different EU countries (e.g., Peng et al. 2011; Wu et al. 2017), but very few address the Indian context (Raman and Aashish 2021). Existing research models and findings from those countries are not readily applicable to the Indian context, as India is a country with multiple cultures and languages and a low literacy rate. Therefore, this study aimed to determine the precursors to Indian retailers' behavioral intention to use mobile payment systems following the COVID-19 pandemic.

Two theories—the theory of planned behavior (TPB) and self-determination theory (SDT)—have been used extensively to investigate behavioral intention (Al-Jubari 2019; Andersen et al. 2000; Deci and Ryan 2012). However, very few studies have integrated SDT and TPB to study behavior intention in-depth (Al-Jubari 2019; Hagger and Chatzisarantis 2009). This study, therefore, exploited an integrated approach by combining the motivational constructs of SDT and the cognitive, attitudinal constructs of TPB to provide a deep understanding of the factors informing the behavioral intention to use mobile payment. In other words, this study attempts to understand (1) how the basic "psychological needs" constructs can explain the creation of cognitive constructs (i.e., attitudes, subjective norms, and perceived behavioral control) generating the behavioral intention to use mobile payment systems for financial transactions; (2) how the integrated approach that combines TPB and SDT helps to clarify, justify and predict the behavioral intention of Indian retail owners to use the mobile payment system following the COVID-19 pandemic.

Based on the above discussion and the underlying research questions, this research will address the following objectives:

- Inspect the role of retailers' basic psychological needs in enhancing their attitudes, subjective norms, perceived control behavior, and intentions to use mobile payment options;
- Scrutinize the interceding effect of the three crucial cognitive constructs (attitude, subjective norms, and perceived behavioral control) on the relationship between basic psychological needs and intention to use mobile payment;
- Predict the behavioral intention of retailers to use mobile payment applications following the COVID-19 pandemic;
- Assess the effectiveness of the hybrid approach (PLS-SEM and ANN) in testing and validating the proposed theoretical framework.

The remainder of the paper is organized as follows. The previous literature related to this study is discussed in Section 2. Section 3 is devoted to a detailed discussion of the theoretical framework that is used in this study. The detailed research methodology is explained in Section 4. The results and discussion are presented in Sections 5 and 6, respectively. The study's implications are presented in Section 7. Finally, the paper offers its conclusion, identifies its limitations, and suggests future research avenues in Section 8.

## 2. Literature Review

### 2.1. Theory of Planned Behavior

The theory of planned behavior has been recognized as a framework to predict an individual's intention to engage in a behavior at a specific place and time. Several other

theoretical frameworks also exist in the literature, but the advantage of TPB has been recognized by several researchers (Al-Jubari 2019; Fayolle et al. 2014; Shin et al. 2018). TPB considers the individual's personal, social, and environmental factors. TPB mainly uses three conceptually independent antecedents—attitudes (AT) toward the behavior, subjective norms (SN), and perceived behavioral control (PBC)—to predict behavioral intention (Liao and Fang 2019; Tommasetti et al. 2018). The attitude (AT) towards a behavior refers to people's overall evaluation (positive or negative) or assessment of the behavior under study (Ajzen 1991). Subjective norms (SN) refer to "a person's beliefs on how and what to think about the people who are considered important motivators to complete the task" (Ajzen 1991). Perceived behavioral control (PBC) can be defined as the factor that "measures the perceived easiness or difficulty in performing the behavior, and it assumes to reflect past experiences as well as anticipates the impediment and obstacles" (Ajzen 1991). PBC factors can be assumed to reflect past experiences and the anticipation of obstacles towards the intention.

TPB models have been successfully adopted in various fields to predict and elucidate a wide range of behavioral intentions (Al-Jubari 2019; Chaulagain et al. 2021; Suen et al. 2020). However, TPB does not differentiate between the beliefs and the assessment of the behavioral outcomes (Al-Jubari 2019; Hagger and Chatzisarantis 2009). That is, TPB does not answer the question "do people engage in a behavior because they choose to or because they are compelled to?" Hence, TPB alone cannot explain the precursors of behavioral intentions. Therefore, SDT was used along with TPB to clarify and justify the precursors to the behavioral intention (Al-Jubari 2019; Andersen et al. 2000; Roca and Gagné 2008).

### 2.2. Self-Determination Theory (SDT)

Self-determination theory (SDT) is primarily related to human motivation. SDT suggests that every individual has motivations for growth and accomplishment. These natural human tendencies can be influenced by different social settings (Ryan and Deci 2000). SDT has three core concepts. First, SDT claims that every individual has three psychological needs: competence, autonomy, and relatedness. All these needs must be met for the individual to function efficiently and mature psychologically (Deci and Ryan 2012). Second, SDT differentiates between intrinsic and extrinsic motivations. Extrinsic motivation involves outcomes, such as pride, prestige, wealth, and continued employment. Intrinsic motivation, on the other hand, reflects an individual's interest and enjoyment. The social environment is the third facet of SDT. The social environment can be observed as being either supportive or unsupportive. The supportive social environment centers on the belief that needs can be satisfied through autonomous actions. In contrast, an unsupportive social environment causes people to feel controlled, resulting in low-quality performance (Deci and Ryan 2012). SDT also contains several different sub-theories. One of these is the basic psychological needs theory (BPNT), which talks about humans' basic need to pursue their behavioral intentions (Chen et al. 2015; Ryan and Deci 2000). These needs (e.g., autonomy, competence, and relatedness) are applicable in all dimensions of human life and universal across people and cultures (Milyavskaya and Koestner 2011). All the above psychological needs influence individuals' behavioral intentions (Milyavskaya and Koestner 2011).

In this study, SDT is integrated with TPB to explain the precursors that predict behavioral intention. To test and validate the above model, robust hybrid PLS-SEM and ANN methods are used.

### 2.3. PLS-Artificial Neural Network Method

The research aimed to assess the hypothesized relationships between constructs and the predictive power of the conceptual model by using a multi-analytical method. The multi-analytical approach integrated an artificial neural network analysis and PLS-SEM modeling. Structural equation modelling (SEM) techniques presume linearity in the relationship, which might oversimplify the analysis of users' behavior intentions (Leong et al. 2013; Qin and McAvoy 1992). On the other hand, artificial intelligence tech-

niques (e.g., neural networks) are suitable for examining linear and nonlinear relationships among variables (Leong et al. 2015). Furthermore, the ANN-based techniques do not require any presuppositions, e.g., normality, homoscedasticity, linearity, or multi-collinearity (Tan et al. 2014), as opposed to traditional statistical methods. Unfortunately, an artificial neural network analysis does not allow causal paths between variables to be statistically assessed (Ahani et al. 2017; Chong 2013). Thus, both ANN and SEM are used complementarily to overcome their respective weaknesses, and this study incorporated both techniques to assess and validate the proposed relationship. First, the PLS framework was used to assess the measurement model and the statistical significance of the hypothesized (causal and mediating) relationships. Second, the significant relation that was obtained by PLS modelling was subjected to an ANN analysis to determine the predictive capacity of the input variables and their relative importance.

The researcher believed that behavioral intention could only be better understood using a proper theoretical framework (Radel et al. 2017). Therefore, a theoretical framework to support the above objectives is proposed. The detailed justification of the proposed theoretical framework, which uses the relevant constructs of TPB and SDT to predict the retailers' intentions to use the mobile payment system, is discussed in the next section.

## 3. Theoretical Framework

The COVID-19-related lockdown measures have facilitated and encouraged the use of digital payment systems in India. The online payment systems benefited the consumers and retailers in different ways. The success and penetration of mobile payment systems largely depend on the retailers' intent to use digital payment platforms. Therefore, to understand the motivation and cognitive processes involved in the behavior affecting the use of mobile payment systems, integrating the theory of motivation (SDT) with the social cognitive theory (TPB) appears to be suitable (Al-Jubari 2019; Hagger and Chatzisarantis 2009). Combining both theories can be helpful to study the intention towards a behavior more deeply because together they are thought to "provide complementary explanations of the processes that underlie motivated behavior" (Chaulagain et al. 2021; Hagger and Chatzisarantis 2009; Li and Wu 2019). The integration of TPB and SDT is not new: previous studies have successfully integrated both theories in various fields, mainly in health and sports studies (Al-Jubari 2019; Hagger and Chatzisarantis 2009; Li and Wu 2019). In their study, Li and Wu (2019) found that the integrated model explained the voluntary intention to continue the act better than using only SDT or TPB. Hagger and Chatzisarantis (2009) have integrated SDT motivational constructs (autonomy, competence, and relatedness) and the perceptual factors of TPB (AT, SN and PBC) to predict behavioral intentions. They argued that integrating SDT and TPB can help to understand the quality of behavioral intention. Barkoukis et al. (2010) and Al-Jubari (2019) observed that the basic psychological need constructs (autonomy, competence, and relatedness) exclusively predict the autonomous intention. A study by Roca and Gagné (2008) tested the applicability of SDT to clarify the role of both intrinsic and extrinsic motivation in the acceptance of e-learning. Their findings showed that when people felt autonomous and competent (i.e., when their basic needs were satisfied), they were more excited to continue using IT.

From the above discussion and the knowledge that was derived from previous literature, the usefulness of both SDT and TPB to predict intent became very apparent. Therefore, this study proposed an integrated model using relevant constructs from both SDT and TPB to understand the precursors to the retailers' intention to use mobile payment systems for their business transactions following the COVID-19 lockdown in India (Figure 1).

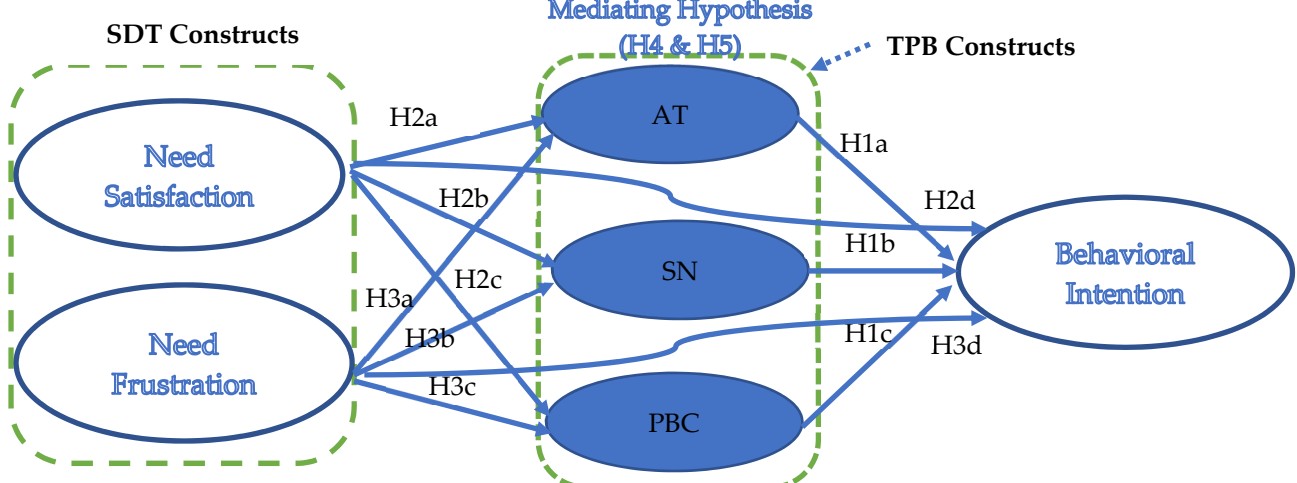

**Figure 1.** Proposed theoretical framework.

The framework proposes three proximal predictive precursors of intent: attitude (AT), social norms (SN), and perceived behavioral control (PBC). However, as discussed, TPB does not distinguish between self-determined outcomes and controlled outcomes to predict intentions (Hagger and Chatzisarantis 2009). Therefore, motivational factors from SDT are used to explain the perpetual constructs of TPB (Al-Jubari 2019; Andersen et al. 2000). The SDT requires constructs such as need satisfaction and frustration to predict TPB constructs and behavioral intention. The need satisfaction and need frustration constructs are derived from individuals' psychological needs (autonomy, competence, and relatedness).

*Hypothesis Development*

In the proposed framework (Figure 1), the relevant constructs of the two complementary theories are taken together to provide an excellent grounding for understanding the antecedents of behavioral intention (Chaulagain et al. 2021; Hagger and Chatzisarantis 2009; Li and Wu 2019). TPB's attitude construct has shown a reliable and robust influence on behavioral intent in several studies in various cultural settings (Almobaireek and Manolova 2012; Chaulagain et al. 2021; Tommasetti et al. 2018; Wu et al. 2017). Ong et al. (2022) also observed that attitude is the major predictor of intention. However, some studies reported contradictory results. Siu and Lo (2013) found that attitude failed to predict an intention to start a new business in a collectivist context.

TPB constructs (e.g., PBC) have shown some differences in predicting intention in various contexts (Almobaireek and Manolova 2012; Moriano et al. 2012). Recently, Liao et al. (2022) found that the direct effects of attitudes and perceived behavioral control on intention are significant. SN has been found to positively influence behavioral intention in several studies (Al-Jubari 2019; Ajzen 1991), but in another study, it was found to influence behavioral intention negatively (Javid et al. 2022). Based on these studies and their conclusions, the following hypothesis is proposed:

**H1.** *The attitude (H1a), subjective norms (H1b), and perceived behavioral control (H1c) significantly predict the retailers' intention to use the mobile payment system.*

As discussed, the need for satisfaction influences intrinsic motivations and the need frustration impacts extrinsic motivations. These two SDT constructs have been found to influence attitude, subjective norms, and perceived behavioral control (Teixeira et al. 2018). Chan et al. (2014) found in their study that social-cognitive factors were significantly associated with intention, and intention to predict participants' reading distance. Widyarini and Gunawan (2017) found that TPB variables, like SDT variables, influence customers' purchase intention. Need satisfaction and need frustration are found to be strong influencers on behavioral intention (Al-Jubari 2019; Hagger and Chatzisarantis 2009). Al-Jubari (2019)

observed that the SDT constructs have an impact on behavioral intention. Hagger and Chatzisarantis (2009) found that motivations are the source of development for AT, SN, and PBC. They found both SDT constructs (NS and NF) played a crucial role in predicting participants' health behavior. Further, Chan et al. (2014) in their research observed that perceived autonomy in SDT significantly predicted attitude, subjective norm, and perceived behavioral control. Further, Chan et al. (2020) in their study established the underlying role of SDT constructs to predict TPB constructs. Thus, the following hypotheses are proposed:

**H2.** *Psychological need satisfaction significantly influences (directly) the attitude (H2a), subjective norms (H2b), perceived behavioral control (H2c), and behavioral intention (H2d).*

**H3.** *Psychological need frustration is significantly linked (directly) to the attitude (H3a), subjective norms (H3b), perceived behavioral control (H3c), and behavioral intention (H3d).*

Various studies have shown the usefulness of integrating SDT and TPB to better understand and differentiate the different precursors to behavioral intention (Al-Jubari et al. 2019; Chaulagain et al. 2021; Hagger and Chatzisarantis 2009; Sicilia et al. 2020; Suen et al. 2020). All these studies have established the mediating role of AT, NS, and PBC in predicting behavioral intention from psychological need constructs. Sicilia et al. (2020) successfully tested the mediating role of TPB constructs to predict intention behavior from SDT constructs. Hagger and Chatzisarantis (2009) found that self-controlled motivation predicts intentions to engage in behavior and is significantly mediated by the TPB constructs (e.g., AT and PBC). Based on the information that was gained from past studies regarding the integration of TPB and SDT and the mediating roles of TPB constructs, the following hypotheses are proposed:

**H4.** *The association between need satisfaction and intention to adopt the mobile payment system is mediated by the attitudes toward using an online payment system (H4a), subjective norms (H4b), and perceived behavioral control (H4c).*

**H5.** *The association between need frustration and intention to use mobile payment systems is mediated by the attitudes toward using online payment systems (H5a), subjective norms (H5b), and perceived behavioral control (H5c).*

## 4. Methodology

This study adopted a dual-stage analysis by integrating SEM and ANN techniques. The dual-stage analysis can help to better understand the theoretical framework in the presence of nonlinear relationships in the data set (Lee et al. 2020). First, the hypotheses that were postulated in the study were tested using PLS-SEM techniques. Second, the relationship that was established by PLS-SEM was validated by adopting ANN-based techniques. The following subsection details the instrument and data collection techniques that were used in this study.

### 4.1. Study Instrument

This study adopted a questionnaire-based data collection method. The questionnaire was divided into three sections. The first part was used to explain the purpose of this study. The second was designed to measure the constructs that were used in this study. Each concept/construct was clearly defined to facilitate correct understanding. The last section of the questionnaire was devoted to collecting the participants' demographic information. All the constructs used by the model were measured by research instruments that were adapted from previous studies. The scales for attitude toward behavior, subjective norm, and perceived behavioral control were adapted from Tommasetti et al. (2018). Attitude toward behavior was measured using a 4-item scale. A 4-item scale measured the subjective norm. Perceived behavioral control was measured using a 3-item scale. The behavioral intention scale was modified from the example in the study by Liao and Fang (2019). In total, three questions were asked about respondents' intention to adopt mobile payment for their business during the COVID-19 lockdown.

The essential psychological need satisfaction and frustration scale were adapted from Chen et al. (2015). The need satisfaction scale consisted of 12 items measuring the satisfying aspects of autonomy, competence, and relatedness. Similarly, the need frustration scale consisted of 12 items measuring the frustrating aspects of autonomy, competence, and relatedness. All the items were measured using a 5-point Likert scale. The detailed construct items are given in Appendix A.

As the scales were primarily adapted from western studies, a pilot study was conducted to assess and improve the questionnaire's reliability in the Indian context. In the pilot test, fifty completed questionnaires were collected from retailers in Nagpur city, India. The reliability scores (Cronbach's alpha) of all the constructs were more than 0.7. Thus, the instrument was found to be reliable and usable for data collection in Indian conditions.

*4.2. Participants*

A cross-sectional survey based on a stratified sampling procedure was used for the data collection. The stratification sampling method can increase the precision of statistical estimates (Creswell and Creswell 2017). The sample size of this study was decided based on the requirements of structural equation modeling (SEM). According to Boomsma and Hoogland (2001), a minimum sample size of 200 is required to minimize the bias in the results in SEM. (Schreiber et al. 2006) recommended a ratio of 10 observations per indicator for a good sample size for SEM. In addition, (Wolf et al. 2013) recommended that the minimum sample size should be at least ten times the number of free parameters in SEM. Considering the above recommendations, it was assumed that a minimum sample size of at least 450 would be sufficient to reduce bias in the study results. Aiming to receive 450 responses, a total of 1200 questionnaires were sent to small-scale retailers in four cities (Nagpur, Raipur, Sambalpur, and Durg) across three central Indian states from August 2020 to May 2021 via social media platforms. Participants' contact details were collected mainly from the Yellow Pages and personal contacts. The questionnaires were proportionally distributed according to the city's population. All participants were informed about the survey objectives and assured that their responses would only be used for research purposes. A total of 616 responses were collected, yielding a response rate of around 51%, which is acceptable in any social science research (Hair et al. 2016). The responses were filtered based on the retailers' monthly transactions (less than 100,000 INR per month). Finally, 496 responses were found to be suitable for data analysis.

The participants' demographic information is shown in Table 1. According to the data, the mean age of the participants is approx. 41 years. Approximately 24% of the participants are women. Most of the participants have educational qualifications at the undergraduate level. In addition, more than 50% of the respondents have more than ten years of experience in the retail business.

**Table 1.** Participants' demographic information.

| Variables | Categories | Number |
|---|---|---|
| Gender | Male | 387 |
| | Female | 109 |
| Age | Below 35 | 198 |
| | Above 35 | 298 |
| Education | Graduate and below | 351 |
| | Above graduation | 145 |
| Number of years in retail business | Ten years and below | 202 |
| | Above ten years | 294 |

## 5. Data Analysis

A structural equation modelling (SEM) technique based on a partial least squares (PLS) regression method was used to test the structural model. PLS-SEM is a robust variance-based second-generation SEM technique (Hair et al. 2016). Furthermore, PLS-SEM can cope with complex structural models and reduce measurement errors (Hair et al. 2016, 2017; Kock and Hadaya 2018). In addition to using several data-screening processes for missing values and outlier detection, a common-method variance test and a nonresponse bias test were used to ensure the data quality.

### 5.1. Nonresponse Bias Test

As previously mentioned, a self-administered instrument was used for the data collection. Therefore, before proceeding further with the data analysis, the author investigated whether any potential nonresponse bias existed in the collected data. An extrapolation method based on Armstrong and Overton's principle was employed to check for potential nonresponse biases (Armstrong and Overton 1977). This approach compared and analyzed both early (100) and late (100) responses to identify any potential variances in their mean values by using a *t*-test. Based on the test results, no significant variance ($t = 9.84$; $p < 0.01$) was observed in the samples' mean values. Therefore, there was no response bias detected in the collected data.

### 5.2. Common-Method Variance Test

In cross-sectional studies, the common-method variance presents a severe problem due to the study instruments' response bias (Hair et al. 2017). Therefore, to address the common-method variance problem in this study, the method devised by Podsakoff (2003) was adopted. A Varimax rotation approach was adopted for all 36 items as one factor. The one-factor test yielded six factors (i.e., NS, NF, AT, SN, PBC, and BI) and converged after five iterations. The total variance explained by the test was 37%, which falls well below the recommended threshold of 50% (Harman 1976). These findings confirm that no common-method variance problem exists in the data.

### 5.3. Model Evaluation

Initially, the first- and second-order confirmatory factor analysis (CFA) was performed to assess the factor binding before testing the proposed model. First-order CFA was applied to the 'need' motivation scales and the TPB scales. Each item was loaded on its respective priority constructs (relatedness for need satisfaction and frustration, autonomy for need satisfaction and frustration, and competence for need satisfaction and frustration, in the context of the need motivation scales and AT, SN, PBC, and BI for the TPB scales). Second, these first-order factors were specified to load on a higher-order model, i.e., NS, NF, AT, SN, PBC, and BI (Table 2). All-fit indices of the model were found to be acceptable ($\chi^2 = 451.891$, df = 324, CFI = 0.969, TLI = 0.987, and RMSEA = 0.039). After factor confirmation, it is recommended to determine the reliability and validity of the constructs (convergent and discriminant) before structural model testing.

**Table 2.** Construct reliability and validity.

| Constructs | No. of Items | Item Loading | Cronbach's Alpha | Average Variance Extracted (AVE) | Composite Reliability (CR) |
|---|---|---|---|---|---|
| Attitude (AT) | 4 | 0.781–0.863 | 0.784 | 0.661 | 0.823 |
| Subjective norms (SN) | 3 | 0.723–0.891 | 0.762 | 0.653 | 0.793 |
| Perceived behavioral control (PBC) | 4 | 0.700–0.853 | 0.783 | 0.641 | 0.824 |
| Need satisfaction (NS) | 12 | 0.683–0.833 | 0.744 | 0.672 | 0.791 |
| Need frustration (NF) | 12 | 0.644–0.811 | 0.745 | 0.630 | 0.782 |
| Behavioral intention (BI) | 3 | 0.713–0.842 | 0.772 | 0.681 | 0.794 |

5.3.1. Construct Reliability and Validity

The construct validity and reliability of the proposed model were evaluated using internal reliability (IR), convergent validity (CV), and discriminant validity (DV). Cronbach's alpha and composite reliability (CR) were used to determine the IR of the constructs. The average variance extracted (AVE) was used to assess the CV of the constructs. The item loading range, Cronbach's alpha, AVE, and CR values are listed in Table 2.

The results that are presented in Table 2 demonstrate the reliability and validity of all the proposed constructs. The estimated construct loadings ranged from 0.644 to 0.891. All of the loadings are higher than the recommended values (Hair et al. 2017). The construct reliability was measured by Cronbach's alpha and the CR score (Table 2). The values of Cronbach's alpha ranged from 0.744 to 0.784, and the values of the CR ranged from 0.782 to 0.824. The Cronbach's alpha and CR values of each construct in the proposed model exceed the recommended cutoff of 0.7 (Hair et al. 2017). This implies that each construct under consideration has exhibited a high level of reliability and consistency. Furthermore, the AVE value of each construct is higher than 0.5 (Hair et al. 2017), thereby establishing a high CV for all the constructs.

Each construct's DV was assessed using two methods (Fornell and Larcker's criterion and the heterotrait-monotrait ratio). First, Fornell and Larcker's criterion was used to assess each construct's DV (Fornell and Larcker 1981). Then, the $\sqrt{}$AVEs of each latent construct was compared with its inter-construct correlation. The $\sqrt{}$AVEs of each construct should have been higher than its correlation with other constructs for a good DV (Hair et al. 2017). This meant that the main diagonal values (Table 3) had to be greater than the off-diagonal values. The results in Table 3 demonstrate that $\sqrt{}$AVEs (illustrated diagonally with bold values) are higher for each construct than the inter-construct correlations. Thus, all the construct DVs were found to be satisfactory.

**Table 3.** (a) Constructs' discriminant validity. (b) Constructs' discriminant validity (HTMT method).

| **(a) Constructs' Discriminant Validity** | | | | | | | | |
|---|---|---|---|---|---|---|---|---|
| Construct | Mean | SD | AT | SN | PBC | NS | NF | BI |
| AT | 3.383 | 0.213 | **0.813** | | | | | |
| SN | 3.235 | 0.561 | 0.467 * | **0.808** | | | | |
| PBC | 2.981 | 0.373 | 0.451 * | 0.341 * | **0.800** | | | |
| NS | 3.211 | 0.712 | 0.419 * | 0.412 * | 0.358 * | **0.819** | | |
| NF | 2.913 | 0.811 | 0.323 * | 0.378 * | 0.322 * | −0.331 * | **0.793** | |
| BI | 3.284 | 0.521 | 0.448 * | 0.433 * | 0.431 * | 0.458 * | 0.381 * | **0.825** |
| **(b) Constructs' Discriminant Validity (HTMT Method)** | | | | | | | | |
| Constructs | AT | | SN | | PBC | NS | | NF |
| SN | 0.482 [0.452, 0.511] | | | | | | | |
| PBC | 0.463 [0.429, 0.478] | | 0.349 [0.322, 0.378] | | | | | |
| NS | 0.411 [0.370, 0.431] | | 0.421 [0.389, 0.449] | | 0.354 [0.334, 0.375] | | | |
| NF | 0.313 [0.281, 0.332] | | 0.389 [0.356, 0.419] | | 0.319 [0.289, 0.343] | 0.324 [0.292, 0.351] | | |
| BI | 0.444 [0.423, 0.474] | | 0.441 [0.414, 0.463] | | 0.433 [0.410, 0.473] | 0.449 [0.415, 0.484] | | 0.391 [0.367, 0.419] |

Note: * $p < 0.05$; attitude (AT); subjective norms (SN); perceived behavioral control (PBC); need satisfaction (NS); need frustration (NF); behavioral intention (BI).

However, researchers have raised concerns about Fornell and Larcker's criterion for DV (Henseler et al. 2009). Therefore, in the second step, the heterotrait-monotrait ratio (HTMT) was used to confirm the previous results (Henseler et al. 2009). HTMT is measured as the average of the heterotrait-hetero correlations of a multitrait-multimethod (MTMM) matrix. The HTMT values of each construct are presented in Table 3. All the HTMT

values were below 0.9. Therefore, all the constructs demonstrated a good DV at a 90% confidence interval (Henseler et al. 2009). Bootstrapping (4999 samples) was used to obtain the $HTMT_{inference}$ interval. Good discriminant validity was assumed when the confidence interval of $HTMT_{inference}$ did not contain "one" (Acosta-Prado et al. 2020; Henseler et al. 2015). The results demonstrated that there was no "one" in the $HTMT_{inference}$ interval (Table 3). Therefore, all the proposed constructs satisfied an appropriate level of DV.

### 5.3.2. Model Assessment

The proposed hypotheses were tested using PLS-SEM. The goodness of fit (GoF), path coefficient, and coefficient of determination ($R^2$) were used to assess the overall quality of the proposed model. The GoF was determined by the geometric mean of the average commonality and average $R^2$ value (GoF $= \sqrt{\text{AVE} * \overline{R^2}}$), as suggested by Alolah et al. (2014). The recommended threshold value of GoF is 0.362 (Alolah et al. 2014). The GoF value for the proposed model was 0.591, which is greater than the threshold values. Therefore, the overall quality of the model was satisfactory. The relationships between the dependent and independent variables were then tested using the path coefficient and t-statistics. Finally, the bootstrapping resampling method was used to determine the coefficients, with the number of iterations fixed at 1000.

The hypotheses concerning the direct relationship between AT, SN, PBC, NS, NF, and BI are shown in Figure 2. Considering AT as a dependent variable, the direct effects of need satisfaction (NS) and need frustration (NF) are significant, with path coefficients of 0.462 and 0.232, respectively. This demonstrates that need satisfaction and frustration positively influence retailers' attitudes toward using mobile payment systems following the COVID-19 pandemic. Hypotheses H1a and H2a are therefore supported. Similarly, both need satisfaction and frustration positively impact the subjective norm (with path coefficients of 0.368 and 0.232, respectively). Hypotheses H1b and H2b are therefore also supported.

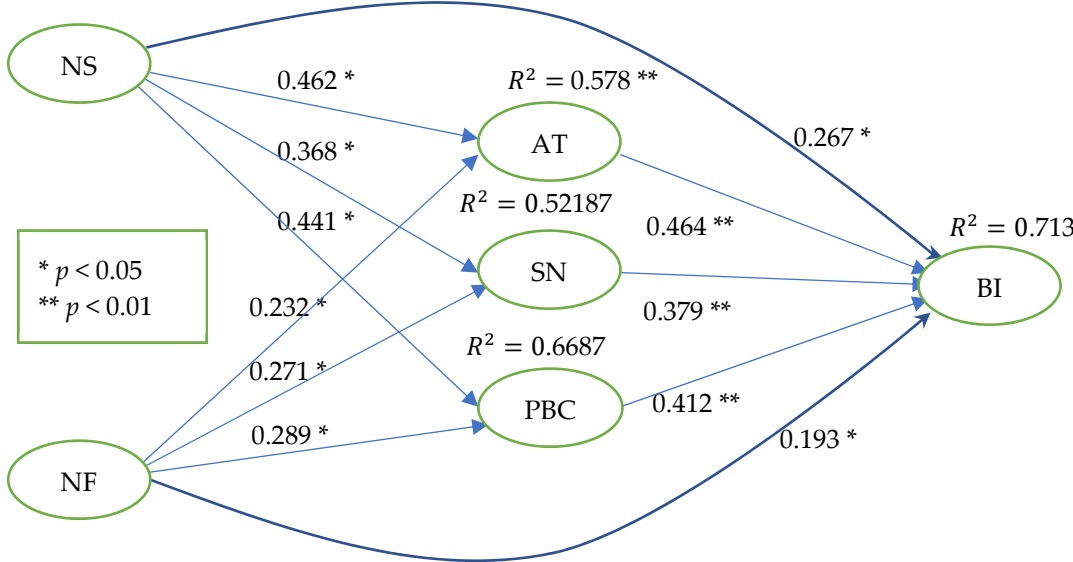

**Figure 2.** Path coefficients (direct).

Need satisfaction (0.441) and need frustration (0.289) positively influenced the perceived behavioral control towards mobile payment systems; therefore, H1c and H2c are supported. The results also show that there is a significant direct effect of need satisfaction (0.267) and need frustration (0.193) on behavioral intention. Hypotheses H2d and H3d are therefore supported. Furthermore, from Figure 2, it is observed that the behavioral intention to use mobile payment systems is positively and significantly influenced by

attitudes to using online payment systems (0.464), subjective norms (0.379), and perceived behavioral control (0.412). Hypotheses H3a, H3b, and H3c are supported, meaning that attitude, subjective norms, and perceived behavioral control are proven positive precursors of behavioral intention to use mobile payment after the COVID-19 pandemic.

The above hypothesis results are confirmed by the direct impact of independent variables on their respective dependent variables. Furthermore, from the above hypothesis test results, it is also observed that motivational factors, such as need satisfaction and need frustration positively influence attitude, subjective norms, perceived behavioral control, and behavioral intention. Hence, SDT constructs (NS and NF) are proved to be precursors to AT, NS, PBC, and BI for adopting mobile payment applications by retail owners following the COVID-19 pandemic.

So far, from the above findings, it is observed that the proposed structural model has significant explanatory power (i.e., the $R^2$ value is 0.713). However, the researchers have ascertained that $R^2$ alone is not sufficient to assess the goodness-of-fit of a structural model in the PLS-SEM approach (Hair et al. 2016). Consequently, the predictive power of the proposed model was estimated using Stone-Geisser's $Q^2$ test (Stone 1974). The $Q^2$ value was calculated by using the blindfolding procedure. $Q^2 \neq 0$ indicates a significant predictive relevance for its endogenous variables (Hair et al. 2017). The $Q^2$ value of the proposed model was found to be 0.374, which indicates a strong predictive relevance for retail business owners' intention to use mobile payment applications following the COVID-19 pandemic. After testing the direct impact of the independent variables on their respective dependent variables, the following sub-section explores the mediating effect of AT, SN, and PBC on the relationship between psychological need constructs (NS and NF) and behavioral intention.

### 5.4. Mediation Effect

One of the most important objectives of this study was to assess the mediating effects of attitude, subjective norms, and perceived behavioral control on the relationship between need constructs (need satisfaction and need frustration) and the behavioral intention to use mobile payment systems in India. The proposed model was complex, with multiple parallel mediations. Therefore, parallel multiple-mediator models, as suggested by Singh et al. (2014) were adopted. The mediation results are presented in Table 4. They confirm the mediating role of AT, SN, and PBC on the relationship between need constructs (NS, NF) and the behavioral intention to use mobile payment. Moreover, all the standard betas (β) are found to be significant. As a result, hypotheses H4 and H5 are supported.

**Table 4.** Indirect effect.

| Path | Standard Beta (β) | t-Value |
|---|---|---|
| NS → AT → BI | 0.354 * | 7.12 |
| NS → SN → BI | 0.189 * | 3.79 |
| NS → PBC → BI | 0.244 * | 4.89 |
| NF → AT → BI | 0.271 * | 5.91 |
| NF → SN → BI | 0.134 * | 2.98 |
| NF → PBC → BI | 0.189 * | 3.81 |

Note: * *p* < 0.01 and critical t-values: 2.58; attitude (AT); subjective norms (SN); perceived behavioral control (PBC); need satisfaction (NS); need frustration (NF); behavioral intention (BI).

Despite the above results (Table 4), even if the mediations are confirmed, proper investigations are required to validate the mediation and assess the size of mediation (full or partial). Therefore, the bootstrapping approach of Preacher and Hayes (2008), along with Sobel's test for mediation (Sobel 1982), was used to assess the types of indirect effects between (NS, NF) and BI, via attitude, subjective norms, and perceived behavioral control. To assess the quality of the mediating effect, the direct effect of (NS, NF) → BI was first assessed without including the mediators. The findings are presented in Table 5 and reveal

that both NS and NF have a significant direct effect on BI without a mediator ($\beta = 0.423$; t-value = 10.456 for NS → BI; $\beta = 0.244$; t-value = 4.90 NF → BI; $p < 0.01$).

**Table 5.** Mediation test results.

| Mediation Analysis (NS → BI) | Standard Beta (β) | SE. | t-Value |
|---|---|---|---|
| **Model 1 (mediation of attitude): NS → AT → BI** | | | |
| NS → AT (a) | 0.491 * | 0.082 | 11.87 |
| AT → BI (b) | 0.511 * | 0.034 | 12.134 |
| NS → BI (c) without mediator | 0.423 * | 0.067 | 10.456 |
| NS → BI (d) with mediator | 0.344 * | 0.114 | 7.091 |
| **Model 2 (mediation of social norms): NS → SN → BI** | | | |
| NS → SN (a) | 0.478 * | 0.089 | 11.564 |
| SN → BI (b) | 0.383 * | 0.077 | 8.129 |
| NS → BI (c) without mediator | 0.423 * | 0.081 | 10.456 |
| NS → BI (d) with mediator | 0.321 * | 0.114 | 6.934 |
| **Model 3 (mediation of perceived behavioral control): NS → PBC → BI** | | | |
| NS → PBC (a) | 0.481 * | 0.054 | 11.567 |
| PBC → BI (b) | 0.472 * | 0.093 | 11.483 |
| NS → BI (c) without mediator | 0.423 * | 0.075 | 10.456 |
| NS → BI (d) with mediator | 0.341 * | 0.092 | 7.090 |
| **Mediation Analysis (NF → BI)** | **Standard Beta (β)** | **SE.** | **t-Value** |
| **Model 4 (mediation of attitude): NF → AT → BI** | | | |
| NF → AT (a) | 0.389 * | 0.095 | 8.24 |
| AT → BI (b) | 0.511 * | 0.082 | 12.1344 |
| NF → BI (c) without mediator | 0.244 * | 0.049 | 4.903 |
| NF → BI (d) with mediator | 0.219 * | 0.064 | 4.713 |
| **Model 5 (mediation of social norms): NF → SN → BI** | | | |
| NF → SN (a) | 0.314 * | 0.101 | 6.911 |
| SN → BI (b) | 0.378 * | 0.041 | 8.191 |
| NF → BI (c) without mediator | 0.244 * | 0.049 | 4.903 |
| NF → BI (d) with mediator | 0.169 * | 0.084 | 3.518 |
| **Model 6 (mediation of perceived behavioral control): NF → PBC → BI** | | | |
| NF → PBC (a) | 0.473 * | 0.078 | 11.479 |
| PBC → BI (b) | 0.467 * | 0.044 | 11.481 |
| NF → BI (c) without mediator | 0.244 * | 0.063 | 4.903 |
| NF → BI (d) with mediator | 0.189 * | 0.089 | 3.812 |
| **Bootstrapping for Specific Indirect Effects (Preacher and Hayes 2008)** | | | |
| **Model** | **Standard Beta (β)** | **t-Value** | |
| Model-1: NS → AT → BI | 0.328 * | 7.021 | |
| Model-2: NS → SN → BI | 0.312 * | 6.124 | |
| Model-3: NS → PBC → BI | 0.324 * | 6.933 | |
| Model-4: NF → AT → BI | 0.234 * | 4.813 | |
| Model-5: NF → SN → BI | 0.151 * | 3.156 | |
| Model-6: NF → PBC → BI | 0.191 * | 3.817 | |

**Table 5.** *Cont.*

| Mediation Analysis (NS → BI) | Standard Beta (β) | SE. | t-Value |
|---|---|---|---|
| Test for mediation (Sobel 1982) | | | |
| **Model** | **z-Values** | | |
| Model-1: NS → AT → BI | 4.100 * | | |
| Model-2: NS → SN → BI | 3.601 * | | |
| Model-3: NS → PBC → BI | 4.642 * | | |
| Model-4: NF → AT → BI | 3.589 * | | |
| Model-5: NF → SN → BI | 2.993 * | | |
| Model-6: NF → PBC → BI | 3.556 * | | |
| Mediation Effects Size (Hair et al. 2017) | | | |
| **Model** | **VAF (Approx)** | **Size** | |
| Model-1: NS → AT → BI | 44 | Partial | |
| Model-2: NS → SN → BI | 36 | Partial | |
| Model-3: NS → PBC → BI | 40 | Partial | |
| Model-4: NF → AT → BI | 47 | Partial | |
| Model-5: NF → SN → BI | 41 | Partial | |
| Model-6: NF → PBC → BI | 54 | Partial | |

Note: * $p < 0.01$ and critical t-values and z-value: 2.58; attitude (AT); subjective norms (SN); perceived behavioral control (PBC); need satisfaction (NS); need frustration (NF); behavioral intention (BI).

The z value of Sobel's test was calculated using Soper's guidelines (Soper 2019). The z value was calculated using the formula $(a \times b / \sqrt{(b^2 \times SE._a^2 + a^2 \times SE._b^2)})$, where SE is the standard error. Variance accounted for (VAF) determined the mediation size effect. VAF was calculated according to the Hair et al. (2017) guidelines using the formula $\%(a \times b / (a \times b + d))$. The 'VAF %' value specifies the ratio of an indirect effect to the total effect. The detailed results are presented in Table 5.

As shown in Table 5, for Model 1 (i.e., NS → AT → BI), need satisfaction is found to be a significant positive predictor of behavioral intention (β = 0.344; t-value = 7.091; $p < 0.01$). In addition, attitude as a proposed mediator has a significant positive impact on BI (β = 0.511; t-value = 12.134; $p < 0.01$). Sobel's test result for the indirect effect of NS on BI is also found to be positive and significant (z = 4.100; $p < 0.01$). Furthermore, the bootstrapping results revealed no 'zero' in the range (lower, upper) of the confidence interval. Hence, all the mediation conditions between NS and BI are satisfied (Farooq and Salam 2020; Preacher and Hayes 2008). This reconfirms hypothesis H4a. Similarly, for Model 2 (i.e., NS → SN → BI), NS is found to have a significant and positive impact on BI (β = 0.324; t-value = 6.933; $p < 0.01$). The proposed mediator subjective norms have a positive and significant impact on BI (β = 0.378; t-value = 8.131; $p < 0.01$). Sobel's test for assessing the indirect effect of NS on BI also found it to be positive and satisfactory (z = 3.601; $p < 0.01$). In addition to the above, the bootstrapping results revealed no 'zero' in the range (lower, upper) of the confidence interval. A significant and positive mediation of SN between NS and BI was found to exist, reconfirming hypothesis H4b. Likewise, for Model 3 (i.e., NS → PBC → BI), NS is positively and significantly linked to BI (β = 0.341; t-value = 7.090; $p < 0.01$). The proposed mediator perceived-behavior control positively and significantly impacted BI (β = 0.467; t-value = 11.481; $p < 0.01$). Sobel's test for assessing the indirect effect of NS on BI was also found to be positive and satisfactory (z = 4.612; $p < 0.01$). The bootstrapping results revealed no 'zero' in the range (lower, upper) of the confidence interval. The significant and positive mediation of PBC between NS and BI was found to be satisfactory, reconfirming hypothesis H4c. Similarly, the impact of mediation of AT, SN, and PBC between need frustration and behavior intention is observed using models (4, 5, and 6), and the results are presented in Table 6. They confirm a significant and positive mediating effect of AT, SN, and PBC on the relationship between NF and BI, reconfirming hypotheses H5a, H5b, and H5c.

**Table 6.** RMSE for the models.

| Network | Model A | | Model B | | Model C | | Model D | |
|---|---|---|---|---|---|---|---|---|
| | Train | Test | Train | Test | Train | Test | Train | Test |
| ANN_1 | 0.107 | 0.098 | 0.106 | 0.100 | 0.111 | 0.101 | 0.121 | 0.102 |
| ANN_2 | 0.110 | 0.101 | 0.106 | 0.102 | 0.111 | 0.110 | 0.126 | 0.106 |
| ANN_3 | 0.113 | 0.099 | 0.110 | 0.098 | 0.109 | 0.106 | 0.125 | 0.106 |
| ANN_4 | 0.107 | 0.105 | 0.107 | 0.097 | 0.110 | 0.111 | 0.101 | 0.107 |
| ANN_5 | 0.109 | 0.110 | 0.110 | 0.112 | 0.100 | 0.108 | 0.117 | 0.108 |
| ANN_6 | 0.112 | 0.111 | 0.109 | 0.110 | 0.112 | 0.108 | 0.108 | 0.112 |
| ANN_7 | 0.106 | 0.109 | 0.108 | 0.105 | 0.109 | 0.111 | 0.119 | 0.105 |
| ANN_8 | 0.111 | 0.106 | 0.109 | 0.110 | 0.107 | 0.109 | 0.112 | 0.108 |
| ANN_9 | 0.109 | 0.099 | 0.112 | 0.107 | 0.110 | 0.099 | 0.122 | 0.109 |
| ANN_10 | 0.113 | 0.109 | 0.104 | 0.105 | 0.111 | 0.107 | 0.108 | 0.105 |
| *Avg* | *0.109* | *0.104* | *0.108* | *0.105* | *0.109* | *0.107* | *0.115* | *0.106* |
| *SD* | *0.003* | *0.005* | *0.002* | *0.005* | *0.004* | *0.003* | *0.008* | *0.003* |

Even though the above results established the significant mediation effect of all the proposed mediators in the multiple parallel mediator models, it is essential to know the actual size of mediation (i.e., full or partial) to analyze the mediation impact. VAF % (i.e., variance accounted for) was therefore used to estimate the size of the indirect effect of the mediator in multiple mediator models (Hair et al. 2017). Generally, the VAF values are grouped into three different categories: VAF < 20 % = no mediation; 20% $\leq$ VAF $\geq$ 80% = partial mediation; and VAF > 80% = complete mediation (Hair et al. 2017). As shown in Table 3 for Model 1, the VAF% value is 58%, which means that the indirect effect explains 58% of the total effect of NS on BI (at least in Model 1). Hence, attitude (AT) partially mediates the relation NS → BI. Similarly, the results presented in Table 6 confirm the partial mediation for all the other models (2, 3, 4, 5, and 6). However, the finding also revealed that the mediation effect of attitude on using mobile payment is more potent than that of SN or PBC. Hence, all the findings that are reported above offer deep insight into retailers' motivational factors towards predicting the intention to use mobile payment applications via attitude, subjective norms, and perceived behavioral control in India following the COVID-19 lockdown.

*5.5. Artificial Neural Network (ANN) Analysis*

The previous sections analyzed the causal and mediating relationships in the structural model using the PLS framework. To further validate the predictors of retail owners' behavioral intention to use mobile payment, the neural network model was integrated with PLS-SEM (Qin and McAvoy 1992). The advantages of the hybrid approach were that it became possible to detect potential nonlinear relationships between the constructs and to determine the importance of each predictor (Ahani et al. 2017). The significant predictors resulting from PLS analysis were used as the inputs to the artificial neural network model. All the inputs to the neural network were normalized using the min–max method (0,1) to boost the model's performance (Negnevitsky 2017). The proposed theoretical framework has four endogenous constructs (i.e., AT, SN, PBC, and BI). Therefore, four different neural network models (A B, C, and D) were used to validate the proposed framework.

As shown in Figure 3, in model A, the output variable is the endogenous construct attitude (AT), and the input variables are psychological need satisfaction (NS) and psychological need frustration (NF). The output variable of model B is subjective norms (SN), and its input variables are NS and NF. Model C has two input variables (NS and NF), and the

output variable is perceived behavioral control (PBC). Model D has five input variables and one output variable, which is behavioral intention (BI).

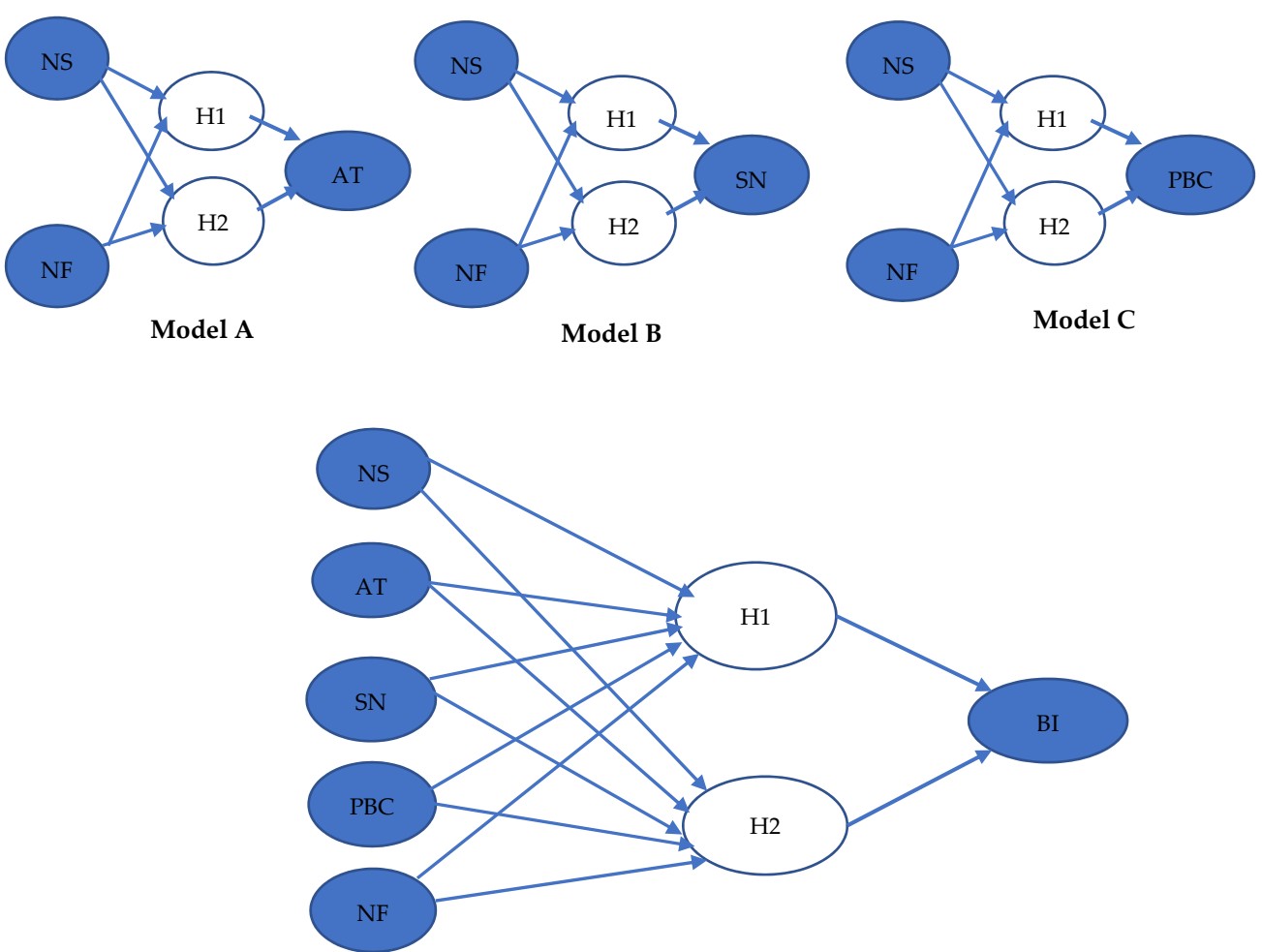

**Figure 3.** Neural network models.

A multilayer perceptron (MLP) backpropagation feedforward algorithm with the sigmoid as the activation function was used to train the model. The MLP is the most popular and widely employed ANN technique (Liébana-Cabanillas et al. 2017; Zabukovšek Sternad S. et al. 2019). The "Neuralnet" R package was used for the experiment (Günther and Fritsch 2010). First, the number of nodes in the hidden layer of each submodel was decided based on two conditions (Liébana-Cabanillas et al. 2017; Zabukovšek Sternad S. et al. 2019): (1) a small number of hidden nodes does not allow complex patterns to be detected; (2) a large number of hidden nodes can trigger overfitting issues. Considering the above conditions, the objective was to find the smallest number of hidden neurons to ensure a suitable generalization of the proposed models (A, B, C, and D). According to the proposition that was put forward by Blum et al. (1997), the optimal number of nodes in the hidden layer should be between the number of inputs and outputs. Based on this proposition, the number of hidden nodes for the model should be between one and two for models A, B, and C. In contrast, it should be between one and five for model D. Next, a trial-and-error procedure was adopted for determining the optimal number of nodes (Chong and Bai 2014; Sharma et al. 2015). Finally, a hidden layer with two nodes was found to be optimal for the proposed models. A 10-fold cross-validation for each model with a data set ratio of 90:10 for training and testing was used to prevent any potential overfitting bias (Liébana-

Cabanillas et al. 2017). The accuracy of the models was assessed by the root-mean-square error (RMSE).

As seen in Table 6, the RMSE values for both the training data and testing data are acceptable for all the models. The low RMSE values in the models ascertain that: (1) the models are efficient and capable of producing high-precision predictions; (2) the parameter estimations are reliable; (3) all input factors are appropriate for predicting the output variables.

The goodness-of-fit coefficient of the ANN models was calculated by adopting the approach that was suggested by Leong et al. (2015). The results show that the input neuron nodes could predict 57.22%, 54.71%, 66.21%, and 72.42% in the variances of AT, SN, PBC, and BI, respectively. The $R^2$ values obtained in the ANN analysis are considerably higher than the values that are obtained in the PLS-SEM analysis (except for model C). This indicates that the dependent (endogenous) constructs in most cases are better explained in the ANN analysis than in the PLS-SEM approach. The superiority in prediction capability in ANN is mainly attributed to the ability of ANN to capture nonlinear relationships. The importance of each input factor in predicting output is determined by the average relative importance and the normalized importance by using Garson's (1991) sensitivity analysis algorithm. The average relative importance was calculated by taking all the ANNs' mean relative importance (ANN1, ANN2... ANN10). The normalized importance of each factor was calculated as the proportion of its relative importance to the factors' maximum relative importance (Leong et al. 2013; Sharma et al. 2015). Table 7 shows the average relative importance and normalized importance of the model and compares the results to the PLS-SEM results.

**Table 7.** Comparison of PLS-SEM and ANN results.

| | ANN Matrix | | | PLS-SEM Matrix | | Comparison |
|---|---|---|---|---|---|---|
| Predictor | Average Relative Importance | Normalized Relative Importance | Ranking | Path Coefficient | Ranking | Matched? |
| | Model A (output: AT) | | | | | |
| NS | 0.771 | 100 | 1 | 0.462 * | 1 | Yes |
| NF | 0.229 | 29.723 | 2 | 0.232 * | 2 | Yes |
| | Model B (output: SN) | | | | | |
| NS | 0.646 | 100 | 1 | 0.368 * | 1 | Yes |
| NF | 0.354 | 54.814 | 2 | 0.271 * | 2 | Yes |
| | Model C (output: PBC) | | | | | |
| NS | 0.721 | 100 | 1 | 0.441 * | 1 | Yes |
| NF | 0.279 | 38.734 | 2 | 0.289 * | 2 | Yes |
| | Model D (output: BI) | | | | | |
| NS | 0.129 | 35.122 | 4 | 0.267 | 4 | Yes |
| NF | 0.104 | 28.334 | 5 | 0.193 | 5 | Yes |
| AT | 0.367 | 100 | 1 | 0.464 | 1 | Yes |
| SN | 0.201 | 54.811 | 2 | 0.379 | 3 | No |
| PBC | 0.199 | 54.223 | 3 | 0.412 | 2 | No |

Note: * $p < 0.05$; attitude (AT); subjective norms (SN); perceived behavioral control (PBC); need satisfaction (NS); need frustration (NF); behavioral intention (BI).

Table 7 illustrates that attitude is the most important predictor of BI to the use of mobile payment services following the COVID-19 pandemic in India. The results also compared the PLS-SEM and ANN analyses regarding the predictor's strength to predict

the outcome. The results show no difference in the importance of the input variable to predict AT, SN, and PBC among the ANN and PLS-SEM approaches. However, there is a difference in the predictive strength of AT, NS, NF, and AT to predict BI. SN and PBC are ranked second and third in the ANN analysis, whereas they ranked third and second in the PLS-SEM analysis. This difference is because ANN computes both the linear and complex nonlinear relationships among predictors with high predictive accuracy, compared to PLS-SEM (Lee et al. 2020).

## 6. Discussion

India has the second-highest number of mobile internet users in the world (after China), and more than 616 million people use the Internet (George and Hatt 2017). However, people generally use the internet to surf social networking sites rather than make mobile payments. Statistics suggest that only 7.6 percent of India's population uses mobile payments for routine transactions (Kats 2018). Recently, the Indian government undertook various initiatives to develop infrastructure through the National Payments Corporation of India (NPCI), which controls all retail payments in the country. This initiative has contributed to the increase in the number of digital payment users in India. COVID-19 restrictions also helped the adoption of online payment systems among potential users. People may shift more towards digital or contactless, cashless payments in the post-pandemic era due to the convenience and rewards of online payment. Therefore, business owners, particularly retailers, should avail themselves of this opportunity by offering digital payment systems to consumers. However, most of the small retailers in India are poor and uneducated, and it is therefore challenging to persuade them to use mobile payment systems. That said, the COVID-19 pandemic-related lockdown forced many small retailers in India to use digital payment platforms. It remains to be seen whether retailers will continue to use mobile payment systems in the post-pandemic era (Franque et al. 2022). Therefore, this study attempted to understand the motivational and social cognitive factors that are responsible for predicting small retail owners' intent to use mobile payment systems following the COVID-19 lockdowns.

A comprehensive, unified model, using well-established social-cognitive and motivational theories (SDT and TPB), was adopted in this study to explore the roles of need motivation and need frustration in facilitating and explaining small retailers' behavioral intention to use mobile payment applications after the COVID-19 lockdown in India. This study also investigates the mediating role of AT, SN, and PBC in the relationship between need factors (NS and NF) and BI. The integration of TPB and SDT helps to provide comprehensive explanations of the role of need motivation in explaining the significance of cognitive constructs (TPB) to predict behavioral intention (Al-Jubari 2019; Hagger and Chatzisarantis 2009; Li and Wu 2019; Roca and Gagné 2008; Suen et al. 2020). This study integrates PLS-SEM with ANN to validate the model and determine each predictor's impact on behavioral intention. The study showed that the two-step approach (PLS-SEM and ANN) offers an in-depth understanding of the proposed theoretical framework. The strength of each predictor of behavioral intention to use mobile payment systems was ranked using an ANN sensitivity analysis to confirm the PLS-SEM findings. The results of the ANN analysis, by and large, confirmed the results that were obtained by SEM. However, there were some differences in the results due to the nonlinear nature and high prediction accuracy of the ANN model.

Overall, the findings that were generated by this research support the argument that need satisfaction and need frustration play a significant role in predicting behavioral intention: 71% of the variance in BI has been explained by the proposed variables (Figure 2). One of the most relevant results to be reported in this study is the confirmation of intrinsic (NS) and extrinsic (NF) motivations as a predictor of mobile payment usage intention during and after the COVID-19 lockdown. The results also show a negative correlation between NS and NF, indicating that NS and NF partially oppose each other. This means

that individuals with high intrinsic motivations toward the business will generally indicate low extrinsic motivations and vice-versa (Al-Jubari 2019).

The intrinsic motivational constructs of SDT (i.e., NS) have significantly influenced the proximal TPB constructs AT, SN, and PBC. Several previous studies reported similar results (Al-Jubari 2019; Hagger and Chatzisarantis 2009). Theoretically, the results of this study have strengthened the justification of SDT and TPB integration and enable the conclusion that the intrinsic motivations could help to enhance retailers' intention to use mobile payment applications even after the COVID-19 pandemic lockdown. Furthermore, as a consequence of intrinsic motivation, there is a greater chance of transformation from intention to action among these retailers. These intrinsically motivated small retail owners will continue to use mobile payments and will accordingly be less likely to abandon the online payment systems. Retailers, whose psychosocial needs are supported by the social and political environment, would favorably perceive and intend to use mobile payment systems after the COVID-19 lockdown due to a sense of promise from the environment and the motivation that originates internally (Al-Jubari 2019). The study results also revealed that the retailers' intention to continue with the mobile payment systems derives from their experience handling different challenging situations during the difficult times of COVID-19, generating confidence to act in the post-COVID-19 world.

In summary, all the hypotheses formulated in this study were supported, highlighting the usefulness of the chosen constructs and their associations in affecting the prediction of behavioral intention to use mobile payment systems. More precisely, the SDT constructs and the mediating role of the TPB construct to predict intention are well supported by previous studies (Fayolle et al. 2014; Li and Wu 2019; Liñán and Chen 2009; Suen et al. 2020).

## 7. Study Implications

### 7.1. Theoretical Implications

The study's findings could lead academicians and practitioners of digital technology, particularly in the banking sector, to consider the proposed framework as a potential tool to investigate the theoretical implications of users' behavioral intention to continue using the digital platform in future pandemic-like situations. This study also makes several other significant contributions to the literature. First, the findings of this study may help to enrich the literature that is related to the precursors of human behavioral intentions in a pandemic situation. Second, this study addressed the need to integrate motivational theory (SDT) and social cognitive theory (TPB) for methodological enhancements. Third, this study supports the arguments that intention can only be better predicted by its proximal cognitive factors, e.g., AT, SN, and PBC (Ajzen 1991; Al-Jubari 2019; Li and Wu 2019; Suen et al. 2020). Fourth, the study also justified using motivational factors (NS and NF) to explain both the intention to continue using a digital platform and the mediating role of AT, SN, and PBC. Fifth, methodologically, this research attempted to overcome the inefficiency of PLS-SEM in handling nonlinear relationships by integrating the ANN model. The proposed hybrid techniques (PLS-SEM and ANN) could help to improve the predictive power of future methodological research paradigms concerning statistical analysis.

### 7.2. Practical Implications

The study could benefit all those who want to understand people's behavioral intentions in an adverse situation like the COVID-19 pandemic and then predict their behavior in post-pandemic situations. This study may help to cultivate a deeper understanding of the motivational and cognitive factors contributing to predicting behavioral intention to use digital payment systems. The empirical findings of this study may also help to guide stakeholders in identifying the critical factors which will help retailers to continue using mobile payment systems. For example, when considering motivation (intrinsic or extrinsic), all necessary steps may be taken to boost their confidence, enhancing their intention to use the digital platform for business transactions. This study has established the importance of motivational factors to predict behavioral intention. Extrinsic motivation focuses

more on instant outputs and may be diluted easily by obstacles and difficulties in system usage. However, in the aftermath of COVID-19 the retailers' extrinsic motivation will probably focus on the external factors (income, sales, rewards, etc.). If they do not receive the expected rewards they will probably discontinue using the mobile payment systems. Therefore, to overcome retailers' frustrations with mobile payment systems, the banking institutions and government should frame policies to attract them. Banks must ensure that all mobile banking services are available 24/7 and hosted on secured platforms. Further, banks should expand their financial services, enhance mobile banking functionalities and maintain the performance, effectiveness, and efficiency of the system. Additionally, banks must establish and maintain relations with retailers to ensure that the system can meet their needs and expectations. Similarly, the government needs to initiate different schemes to attract retailers to use digital platforms for their business transactions. In the recent past, the Indian government has launched various initiatives, such as the DIGIDHAN Mission and Bharat Interface for Money (BHIM) to facilitate fast, secure, and reliable cashless payments through mobile devices. The Indian government's DIGIDHAN Mission aims to improve online infrastructure and increase internet access in rural India. These initiatives can boost Indian retailers' satisfaction and reduce their frustration, resulting in their continued use of mobile payment systems in the future.

Finally, the study results can provide insights into the different ways that retailers can capture new business opportunities by using mobile payment services to adapt to changing consumer behaviors arising from the COVID-19 pandemic. Retailers should develop effective innovation strategies that are appropriate to their products and services to enhance customers' shopping experience and encourage the use of digital payment systems.

## 8. Conclusions

In conclusion, this study adopted an integrated theoretical framework using appropriate cognitive constructs and 'need' motivational constructs to understand retailers' intention to use mobile payment systems following the COVID-19 lockdown in India. Furthermore, the integration of PLS-SEM and ANN as a tool makes it possible to achieve two objectives. First, PLS-SEM helps to effectively test all the formulated hypotheses in a complex theoretical model involving simultaneous parallel mediation. Second, the ANN helps to validate the relationship between endogenous and exogenous variables, even in the presence of nonlinear relationships among datasets (Hew et al. 2019; Lau et al. 2021; Leong et al. 2015). The proposed theoretical model in this study was empirically evaluated using a relatively large sample of around 500 valid responses, which makes the findings generalizable. The findings of this study justified the integration of SDT and TPB models to predict the behavior intention to use mobile banking during and aftermath of the COVID-19 pandemic. In particular, the findings that were generated by this study would be helpful for government policymakers and other agencies as they seek to implement policies that can help retail business owners to use digital banking platforms for their businesses.

### Limitations and Future Scope

Finally, it is worth mentioning that there are limitations to the study, as well as future scope. First, even though the questionnaire helped in data collection and analysis, it did not allow us to understand more comprehensively why negative motivational factors, such as need frustration have a strong positive association with intention. Therefore, a qualitative research-based approach (such as in-depth interviews) may be required to overcome this limitation. Second, the current study sample was restricted to central India, and the motivational factors that were considered in this study (need satisfaction or frustration) are mainly related to people's social expectations. Hence, different cultures and individualities are likely to influence these motivational factors. Therefore, future research should replicate this study in other parts of India or even in other countries. Third, future studies should also consider a multi-group analysis involving different social and demographic characteristics (e.g., the size of the business, gender, age, experience, etc.) to complement and consolidate

the findings further. Fourth, for a better understanding and generalization of the study results, longitudinal data can be considered in future research. Finally, future studies in different domains and different cultures are required to further strengthen the need for the integration of SDT and TPB theories to predict behavior intention.

**Funding:** This research received no external funding.

**Institutional Review Board Statement:** Not applicable.

**Informed Consent Statement:** Not applicable.

**Data Availability Statement:** Data can be available on request.

**Conflicts of Interest:** The authors declare no conflict of interest.

## Appendix A. Study Constructs

| | Variables | Items |
|---|---|---|
| **Attitude (AT)** | AT1 | Adopting a mobile payment system in business transactions is not a good idea during COVID-19. |
| | AT2 | Adopting a mobile payment system in business transactions is a good idea during COVID-19. |
| | AT3 | I like to use a mobile payment system in my business during COVID-19. |
| | AT4 | Consumers who use mobile payment systems adopt appropriate behavior in a pandemic situation. |
| **Subjective norms (SN)** | SN1 | People who influence my behavior think it is preferable to use a mobile payment system during COVID-19. |
| | SN2 | People important to me think it is preferable not to use a mobile payment system during COVID-19. |
| | SN3 | People in my surrounding (friends, family, suppliers, consumers, etc.) consider it useless to use a mobile payment system during COVID-19. |
| | SN4 | In general, people in my surrounding (family, friends, suppliers, consumers, etc.) acknowledge the use of mobile payment systems during COVID-19. |
| **Perceived behavioral control (PBC)** | PBC1 | I would like to adopt a mobile payment system in my retail business during and post- COVID-19 lockdown. |
| | PBC2 | I decide to use mobile payment during and after the post- COVID-19 lockdown. |
| | PBC3 | I think I have the resources, the knowledge, and the skills necessary to adopt mobile payment during and after the COVID-19 lockdown. |
| **Behavioral intention (BI)** | BI1 | I would not like to use a mobile payment system during COVID-19. |
| | BI2 | I intend to use a mobile payment system during and after COVID-19. |
| | BI3 | I cannot say positive things about using the mobile payment system during COVID-19. |
| **Need satisfaction (NS)** | NS1 | I feel a sense of choice and freedom in using mobile payment systems during the COVID-19 pandemic. |
| | NS2 | I feel that my decisions to use mobile payment reflect what I want during the COVID-19 pandemic. |
| | NS3 | I feel my choices express who I am in using a mobile payment system. |

| Variables | Items |
| --- | --- |
| NS4 | I feel I have been doing what really interests me in using mobile banking during the COVID-19 pandemic. |
| NS5 | I feel that the people I care about also care about me using mobile payment during the COVID-19 pandemic. |
| NS6 | I feel connected with people who care for me, and for whom I care in using mobile payment during COVID-19. |
| NS7 | I feel close and connected with other people who are important to me in using mobile payment during COVID-19. |
| NS8 | I experience a warm feeling with the people I spend time with in using mobile payment during COVID-19. |
| NS9 | I feel confident that I can do things well in using mobile payment during COVID-19. |
| NS10 | I feel capable of what I do in using mobile payment during COVID-19. |
| NS11 | I feel competent to achieve my goals in using mobile payment during COVID-19. |
| NS12 | I feel I can successfully complete difficult tasks using mobile payment during COVID-19. |
| NF1 | Most of the things I do feel like "I have to" in using mobile payment during COVID-19. |
| NF2 | I feel forced to do many things I wouldn't choose to do in using mobile payment during COVID-19. |
| NF3 | I feel pressured to do too many things by using mobile payment during COVID-19. |
| NF4 | My daily activities feel like a chain of obligations in using mobile payment during COVID-19. |
| NF5 | I feel excluded from the group I want to belong to in using mobile payment during COVID-19. |
| NF6 | I feel that people who are important to me are cold and distant towards me in using mobile payment during COVID-19. |
| Need frustration (NF)    NF7 | I have the impression that people I spend time with dislike me for using mobile payment during COVID-19. |
| NF8 | I feel the relationships I have are just superficial in using mobile payment during COVID-19. |
| NF9 | I have serious doubts about whether I can do things well in using mobile payment during COVID-19. |
| NF10 | I feel disappointed with many of my performances in using mobile payment during COVID-19. |
| NF11 | I feel insecure about my ability in using mobile payment during COVID-19. |
| NF12 | I feel like a failure because of the mistakes I make in using mobile payment during COVID-19. |

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
