# Peer review of "Investigating and Predicting Intentions to Continue Using Mobile Payment Platforms after the COVID-19 Pandemic: An Empirical Study among Retailers in India"

_jrfm, doi:10.3390/jrfm15070314_

Round 1

Reviewer 1 Report

The title of this paper is quite interesting. However, the theoretical framework of this paper is quite weak. A lot of sections are not explained well. Based on a lot of flaws, I would not recommend that the current manuscript be considered for publication in Journal of Risk and Financial Management.

  • There is too much information in the abstraction section. Try to shorten the abstract section.
  • In the introduction section, I cannot relate the research problems closely with the research objectives of this paper.
  • Why is the "BOPIS" used in this paper? It seems that "BOPIS" is not used in the following sections of this paper.
  • Do not use American and British English spelling interchangeably. Spelling must be consistent throughout this paper.
  • Please try to provide the full name of the COVID-19 for the first time.
  • I just would like to know the section before Section 2.1 is necessary to exist in this paper.
  • I just would like to know the full name of theory of planned behavior needs to be mentioned again since it has been mentioned in the introduction section. Please see the subtitle of Section 2.1.
  • I just do not understand why the research objections are mentioned in Section 2.3.
  • I cannot relate the literature closely with all of the proposed hypotheses.
  • I cannot clearly see how the data were collected.
  • Is it possible to see the respondents' marital status and income in Table 1?
  • Is it possible to see the measurement items for each construct examined in this paper?
  • I cannot see the study variable correlations using a table.
  • I do not see any analyses related to the measurement and structural models.
  • The study implication section should be extended more.
  • Is it possible to divide the conclusion and future scope section into different sections?
  • What is the potential contribution of this paper?
  • Is it possible to cite more recent papers related to this study?
  • Professional proofreading is highly required since there are several grammatical and technical errors throughout this paper?

Author Response

All the suggestions are address in the attached files

Reviewer 2 Report

In-text citations are not aligned. Moreover, some formatting and spacing issues need to be resolved throughout the manuscript.

Please explain how the author conducted pilot testing and how he checked the initial reliability of the questionnaire.

Provide Justification for sample size.

What were the research design and sampling process?

Provide validity through both Fornell and larker and HTMT criteria.

The conclusion and discussion sections need to be revised.

Theoretical contributions and managerial implications sections need to be revised.

Provide recommendations for the retailers based on your findings.

Provide a separate section for limitations and scope for future research.

Moreover, professional proofreading is required for a better understanding of the manuscript.

Author Response

I tried to address all the comments n the attached file 

Round 2

Reviewer 1 Report

I do appreciate the author(s) for their great effort to revise this paper based on my previous comments. It looks much better. However, something still needs to be improved a lot.

* The study implication section needs to be improved. Try to divide the study implication section into the theoretical and practical implication sections.

* There are still grammatical and technical errors throughout this paper. I strongly recommend that the current manuscript MUST be proofread by a native English speaker.

Author Response

All the suggested corrections have been incorporated.

Regards

Reviewer 2 Report

The manuscript has been substantially improved. However, the following points need to be revised before the manuscript can be accepted.

Why does the author apply bootstrapping resampling method with the number of iterations fixed at 1,000 when 5,000 is more reliable and provides consistent results?

Formation and spacing issues throughout the manuscript need to be resolved.

The author has provided all the values, i.e., Loadings, CR, AVE, etc., in two digits; however, the author must include all the values up to three-digit.

All the tables should include notes regarding full forms of constructs and significance values.

In-text references need to be aligned as in some cases, the author used (&), and in a few cases, he used (and).

The study Implications section needs to be discussed in two sections, theoretical implications, and practical implications, for better understanding.

Implications need to be revised based on the results of the study. For example, the author must explain implications only related to the constructs of the present study.

The conclusion section looks more like theoretical implications, so the author must revise this section, and he must only conclude study results.

The limitations and future scope further need to be enhanced, and the author must include at least five solid points for the same.

Author Response

All the suggested changes have been incorporated.

Regards

Round 3

Reviewer 1 Report

I do appreciate the author(s) for their great effort into revising this paper based on my previous comments. It looks much better than before.  I would recommend that the current manuscript be considered for publication in Journal of Risk and Financial Management. Congratulations! Well done!